# Cardiovascular, Metabolic and Inflammatory Changes after Ovariectomy and Estradiol Substitution in Hereditary Hypertriglyceridemic Rats

**DOI:** 10.3390/ijms23052825

**Published:** 2022-03-04

**Authors:** Jan Pitha, Martina Huttl, Hana Malinska, Denisa Miklankova, Hana Bartuskova, Tomas Hlinka, Irena Markova

**Affiliations:** 1Cardiology Department, Cardiac Centre, Institute for Clinical and Experimental Medicine, 140 21 Prague, Czech Republic; 2Laboratory for Diabetes Pathology, Experimental Medicine Centre, Institute for Clinical and Experimental Medicine, 140 21 Prague, Czech Republic; mabw@ikem.cz (M.H.); haml@ikem.cz (H.M.); mild@ikem.cz (D.M.); irma@ikem.cz (I.M.); 3Atherosclerosis Research Laboratory, Experimental Medicine Centre, Institute for Clinical and Experimental Medicine, 140 21 Prague, Czech Republic; kubh@ikem.cz (H.B.); hlit@ikem.cz (T.H.); 4Department of Physiology, Faculty of Science, Charles University, 128 44 Prague, Czech Republic; 5Internal Department, 2nd Medical Faculty, Charles University Prague, 150 06 Prague, Czech Republic; 6Faculty Hospital Motol Prague, 150 06 Prague, Czech Republic

**Keywords:** ovariectomy, cardiovascular changes, estradiol substitution, insulin resistance, hereditary hypertriglyceridemic rat

## Abstract

Background: If menopause is really independent risk factor for cardiovascular disease is still under debate. We studied if ovariectomy in the model of insulin resistance causes cardiovascular changes, to what extent are these changes reversible by estradiol substitution and if they are accompanied by changes in other organs and tissues. Methods: Hereditary hypertriglyceridemic female rats were divided into three groups: ovariectomized at 8th week (*n* = 6), ovariectomized with 17-β estradiol substitution (*n* = 6), and the sham group (*n* = 5). The strain of abdominal aorta measured by ultrasound, expression of vascular genes, weight and content of myocardium and also non-cardiac parameters were analyzed. Results: After ovariectomy, the strain of abdominal aorta, expression of nitric oxide synthase in abdominal aorta, relative weight of myocardium and of the left ventricle and circulating interleukin-6 decreased; these changes were reversed by estradiol substitution. Interestingly, the content of triglycerides in myocardium did not change after ovariectomy, but significantly increased after estradiol substitution while adiposity index did not change after ovariectomy, but significantly decreased after estradiol substitution. Conclusion: Vascular and cardiac parameters under study differed in their response to ovariectomy and estradiol substitution. This indicates different effects of ovariectomy and estradiol on different cardiovascular but also extracardiac structures.

## 1. Introduction

It is still matter of debate whether menopause is an independent cardiovascular risk factor. However, cardiovascular disease (CVD) caused mainly by atherosclerosis is the primary and underscored cause of morbidity and mortality in women [1,2]. Moreover, in addition to traditional risk factors for atherosclerotic disease and subsequent cardiovascular events, insulin resistance associated with metabolic syndrome was definitely proved to be risk factor for atherosclerosis and menopause could further aggravate and, potentially, also even trigger unfavorable vascular changes. In addition, if female sex hormone substitution therapy after menopause could reverse unfavorable vascular changes remains very controversial topic which is intensively investigated on experimental and human level [3,4,5]. Several studies indicated that if hormonal substitution therapy is started early after menopause, it could have favorable vascular effects. This is in contrast to previous unsuccessful studies which initiated hormone substitution therapy later after menopause [6,7]. However, despite many studies focused on the timing of hormone substitution therapy after menopause, less data is available regarding the effect of such intervention on traditional atherosclerotic risk factors and other cardiovascular parameters but especially few data exist regarding effect of hormone substitution therapy on extracardiac organs and tissues. In particular, there are only few data available, to what extent exactly are atherosclerotic and, in general, cardiovascular changes reversible by hormone substitution therapy, which factors could be associated with (un)favorable changes of involved organs and if these changes are associated with changes in other organs and tissues. This topic is of high importance, because the presence of metabolic and other atherosclerotic and cardiovascular risk factors before menopause could strongly determine success of hormonal substitution therapy after menopause. Such interaction between hormone substitution therapy and presence of risk factors or already present vascular pathology could potentially determine future development of CVD in women [8]. Regarding optimal timing of potential intervention, in our previous cross-sectional study of population sample of middle-aged women, we detected much stronger impact of smoking on preclinical atherosclerosis defined as carotid intima media thickness in perimenopausal women than in pre- and postmenopausal women [9], potentially partly mediated by remnant lipoproteins [10]. In addition, in our experimental study in hamster model [11] the hypolipidemic treatment with simvastatin substantially decreased the prevalence of atherosclerotic changes after ovariectomy, but otherwise did not change concentration of atherogenic LDL cholesterol. Nevertheless, statin treatment improved proportions of pro- and antiatherogenic serum lipids by the increase of HDL cholesterol. Therefore, antiatherogenic effect of statin treatment in this study was not mediated by decrease of concentration of LDL cholesterol, but by the increase of concentration of HDL cholesterol. Notably, in this model, the timing of simvastatin treatment had no significant effect on prevention of atherosclerotic changes or lipid parameters. More complex changes after menopause beyond circulating lipoproteins should be, therefore, studied.

In general, development of cardiovascular disease after menopause is a complex and challenging process and the research in this field is focused on unfavorable arterial changes at early stages of cardiovascular disease. Recently the use of non-invasive imaging strategies, such as ultrasound studies of cardiac, but also aortic, cerebral, and peripheral vascular disease models in rodents is widely implemented and ultrasound detection of vascular disease is expanding methodology which allows longitudinal studies of different vascular and cardiac structures in living animals [12]. This could be specifically applicable for studying the effect of menopause on cardiovascular structures prospectively also in animal models.

In the recent study, we focused on the more complex effect of ovariectomy and estradiol substitution on vascular, cardiac, but also extravascular and extracardiac organs and tissues, and metabolic, genetic and inflammatory parameters in already well-established experimental model of insulin resistance, dyslipidemia, mild hypertension and low-grade inflammation, particularly in hereditary hypertriglyceridemic (HHTg) female rats [13].

The main aim was to investigate cardiovascular, metabolic and inflammatory changes caused by ovariectomy in the terrain of insulin resistance, i.e., mimicking menopause in women with already present metabolic and inflammatory burden. We also studied to what extent are these changes reversible with female sex hormone substitution, namely with estradiol. In addition, we analyzed vascular parameters through repeated examination of vascular system by ultrasound, but also structure and content of extravascular and extracardiac organs and tissues.

## 2. Results

At the start of the experiments, study groups (sham, ovariectomized and ovariectomized with estradiol substitution) did not differ in weight and, no pathological changes in abdominal aorta were detected by ultrasound before start of the study. Body weight increased in ovariectomized group during the study and remained stable in the group with estradiol substitution. In addition, food intake increased in ovariectomized females without estradiol substitution compared to other two groups.

### 2.1. Changes of Cardiovascular Parameters after Ovariectomy and Estradiol Substitution 

At the end of the study, the strain of suprarenal aorta after ovariectomy was non-significantly lower in suprarenal and significantly lower in infrarenal aorta and in ovariectomized females with estradiol substitution the strain of suprarenal and infrarenal aorta was significantly higher than in ovariectomized females without estradiol substitution; in suprarenal aorta, the strain after estradiol substitution was higher not only than in ovariectomized rats (*p* = 0.002) but also than in the sham group (*p* = 0.02) (Figure 1, Table A1/Appendix A); even more pronounced increase was detected for the strain in suprarenal aorta, but it was partly caused by higher diameter of this segment measured at the start of the study (data not shown). Similar pattern was found for heart rate. The heart rate at baseline was 297.0 ± 18.0, 324.7 ± 30.0, and 372.0 ± 22.5 beats per minute in sham, ovariectomized group and in ovariectomized group with estradiol substitution, respectively (one-way ANOVA: *p* < 0.001). The heart rate at the end of the study was 319.4 ± 29.7, 284.2 ± 25.9, and 285.0 ± 30.3 beats per minute in sham, ovariectomized group and in ovariectomized group with estradiol substitution, respectively (one-way ANOVA: *p* < 0.881). This means that in ovariectomized rats and in ovariectomized rats with estradiol substitution heart rate significant decreased (Figure A1/Appendix A).

In addition, the expression of the gene for nitric oxide synthase 3 (*Nos3*) measured in the whole abdominal aorta irrespectively of supra/infrarenal segment was significantly lower after ovariectomy while in ovariectomized females treated with estradiol it was similar to the sham group. No significant between-group differences were observed for connexin 37 (*Cx37*) gene expression measured also in the whole abdominal aorta (Figure 2). No between group differences were detected for circulating nitric oxide synthase (NOS) (Table A1/Appendix A). The correlation between *Nos3* and *Cx37* gene expression was r = 0.459 in the whole group, r = 0.752, r = 0.183, and r = 0.624 for the sham group, ovariectomized rats, and ovariectomized rats with estradiol substitution, respectively.

Regarding cardiac parameters, relative weight of myocardium and the weight of the left ventricle were significantly lower in ovariectomized females than in the sham group at the end of the experiment while in ovariectomized females with estradiol substitution they were similar to the sham group. Different pattern was observed for the content of triglycerides in the myocardium which was similar in ovariectomized females at the end of the study as in the sham group, but significantly higher in ovariectomized females with estradiol substitution than in other two groups (Figure 3). However, no between group differences were found in the expression of genes potentially responsible for triglyceride and/or other lipid content in the heart, oxidative stress and other pathophysiological pathways/homeostatic mechanisms: namely lipoprotein lipase, stearoyl-CoA-desaturase-1, nuclear factor E2-related factor 2 and connexin 43 (data not shown).

### 2.2. Changes in Other Organs after Ovariectomy and Estradiol Substitution 

Changes of these parameters are shown in Table 1. Body weight was significantly higher in ovariectomized females than in the sham group, and similar in ovariectomized females with estradiol substitution as in the sham group at the end of the study. In addition, relative and absolute weight of uterus was lower in ovariectomized females than in the sham group, and similar in ovariectomized females with estradiol substitution as in the sham group. Interestingly, the adiposity index was similar in ovariectomized females as in the sham group, but in ovariectomized females substituted with estradiol it was significantly lower than in other two groups; in other words, reversed pattern compared to the content of triglycerides in myocardial tissue was observed. Relative liver weight was significantly lower in ovariectomized females than in the sham group and was found to be higher in ovariectomized rats with estradiol substitution than in the sham group. Similar patterns for the effect of ovariectomy and ovariectomy with estradiol substitution were found for: increased and decreased content of triglycerides in the liver tissue, increased and decreased content of cholesterol in the liver tissue, decreased and increased relative weight of kidneys, and increased and decreased content of triglycerides in the renal cortex, respectively. No between group differences were detected for the content of triglycerides in the skeletal muscle.

### 2.3. Circulating Metabolic, Inflammatory and Hormonal Parameters after Ovariectomy and Estradiol Substitution

Serum estradiol was significantly lower in ovariectomized females than in the sham group, and was significantly higher in ovariectomized females with estradiol substitution, in the latter group reaching very high values. Progesterone was significantly lower in ovariectomized females than in the sham group and remained at the same level in ovariectomized females substituted with estradiol as in ovariectomized females without estradiol substitution. No between group differences were detected for circulating Anti-Mullerian hormone (AMH). Regarding serum cholesterol, no differences were found between study groups. Serum triglycerides were significantly lower in the ovariectomized females than in the sham group while in the ovariectomized females with estradiol substitution they were at the same level as in the sham group. Serum HDL cholesterol was significantly higher in ovariectomized females than in the sham group, and in ovariectomized females with estradiol substitution it was higher than in other two groups. No between group differences were detected for circulating free fatty acids (FFA), non-fasting glucose, insulin and glucagon. The concentration of liver enzymes (AST, ALT) was higher in ovariectomized females than in the sham group, and in ovariectomized females after estradiol substitution it stayed at similar level as in ovariectomized rats. Serum concentration of interleukin 6 (IL-6) was significantly lower in ovariectomized females than in the sham group, and was significantly higher in ovariectomized females with estradiol substitution, in the latter group, the concentration of IL-6 was even higher than in the sham group (*p* = 0.001).

### 2.4. Summary of Changes of Parameters with Regard to Ovariectomy and Estradiol Substitution

Significant changes after ovariectomy reversed by estradiol substitution were increased body weight, decreased expression of *Nos3* in the abdominal aorta, decreased relative weight of myocardium, decreased relative weight of the left ventricle, and changes in the liver and kidney parameters including decreased relative organ weight, increased content of cholesterol and triglycerides in particular organs and tissues, and decreased circulating triglycerides. Significant changes after ovariectomy modified by estradiol substitution to the levels significantly different than were in the sham group was decreased strain of infrarenal segment of the abdominal aorta, increased serum HDL cholesterol and decreased circulating IL-6. Significant changes after ovariectomy not reversed by estradiol substitution were decreased serum progesterone level and increased liver enzymes. No effect of ovariectomy and of the ovariectomy followed by estradiol substitution was observed for AMH, NOS, expression of *Cx37*, circulating FFA, non-fasting glycemia, serum insulin, and glucagon. No effect of ovariectomy but significant changes after ovariectomy with estradiol substitution were observed for the increased triglycerides content in myocardium, increased strain of suprarenal aorta, and decreased adiposity index. 

## 3. Discussion

Regarding vascular parameters under study, the main finding was that aortic strain moderately decreased after ovariectomy but increased significantly if ovariectomy was followed by estradiol substitution; in the suprarenal segment estradiol substitution after ovariectomy led even to significantly higher levels of the aortic strain than in the sham group (Figure 1, Table A1/Appendix A). These vascular changes were accompanied by changes in the expression of *Nos3* gene which demonstrated similar pattern; it decreased after ovariectomy and increased to the levels similar as in the sham group if ovariectomy was followed by estradiol substitution. On one hand, these findings demonstrated unfavorable effect of decrease of sex female hormones, on the other hand, favorable effect of estradiol substitution on the vasculature already was detected as in already detected in human and experimental studies [14,15,16,17]. Moreover, our findings indicate mechanisms potentially more or less involved in the vascular effects of sex hormones, in this case estradiol. In contrast to the favorable effect of estradiol on aortic strain and expression of *Nos3*, there was no effect of the ovariectomy and/or estradiol substitution on the expression of also “cardiovascular” gene for *Cx37* [18,19], which is potential target for intervention in cardiovascular disease [20] and described as important for cardiovascular disease in previous (but not all) studies [21,22,23,24]. The explanation for this negative finding could be that the change of *Cx37* gene expression was not yet detectable relatively soon after ovariectomy and could be relatively delayed compared to the change of the expression of gene for *Nos3*. In addition, changes of *Cx37* activity and gap junctions were described mainly in the smaller vessels including mice and dependent on changes of shear stress [25,26]. Therefore, in abdominal aorta, the changes of shear stress in rats compared to mice could be less pronounced in this location due simply to larger diameter and in contrast to *Nos3* gene, *Cx37* gene could be less sensitive to these changes. Another explanation comes from our pilot experimental study in which the expression of *Cx37* gene was significantly lower in the suprarenal than in the infrarenal segment in four Prague hypercholesterolemic females on standard diet (suprarenal segment of abdominal aorta: 1.009 ± 0.152 vs. infrarenal segment: 2.011 ± 0.297; *p* = 0.015; unpublished data). Notably, it was not observed on hypercholesterolemic diet. Therefore, if we can suppose similar situation in HHTg rats in which we measured *Cx37* gene expression in the whole abdominal aorta, we could miss segment specific differences in expression of particular genes including gene for *Cx37*. We noted, that changes after ovariectomy/estradiol substitution both in *Cx37* gene expression, but also in circulating NOS were similar to changes of aortic strain and of *Nos3* but were not statistically significant (Figure 2, Table A1/Appendix A). In this particular study, Cx37 was involved in basal nitric oxide release, release of cyclooxygenase products and the regulation of the sensitivity for acetylcholine in contrast to connexin 40. We found rather strong correlation between *Nos3* and *Cx37* gene expression which was not present in the ovariectomized group in contrast to the sham group and ovariectomized group with estradiol substitution, but because of relatively low numbers of animals in each group we are cautious to interpret these data on biological grounds.

Regarding cardiac parameters, the relative weight of myocardium and of the left ventricle decreased significantly after ovariectomy and were completely reversed by estradiol substitution to values similar as in the sham group. These findings could be explained by rapid decrease of myocardial mass due also at least partly to rapid decrease of sex hormones and decreased number of estradiol receptors in cardiac tissue after ovariectomy and their restoration after estradiol substitution [27,28]. Changes levels of estradiol could be also the cause of the significant changes of the heart rate, this could be caused also by changes in estradiol receptors function and their impact on sensitivity of the heart to stress. In addition, in recent study it has been shown in mice that cellular content of the heart is dependent on and can be rapidly changed by endocrine factors, particularly gonadal hormones [28,29,30,31]. In previous study estrogen (and also testosterone) were shown to have direct interactions with ion channels and were able to transcriptionally alter ion channel expression in a regionally dependent manner, in rabbit heart [32]. Changes of sex hormone could, therefore, lead to different stress responses of the heart, but it should be noted, that estradiol can have discordant cardiac effects based on the experimental models used. However, in our study rather the rate of change of estradiol concentration was of importance, because similar pattern, decrease of heart rate, was detected both in ovariectomized females and in ovariectomized females with estradiol substitution. Nevertheless, because of different values of heart rates detected at the beginning of the study in our study groups, we are also cautious in the interpretation of these data. Moreover, no between group differences were found in myocardial tissue for the expression of genes which could be potentially responsible for impaired homeostatis especially in lipid metabolism.

In contrast to changes of myocardial mass, content of triglycerides in myocardium changed after ovariectomy differently; the content of triglycerides in myocardium was not affected by ovariectomy but it increased significantly if ovariectomy was followed by estradiol substitution even above the values in the sham group. Increased content of triglycerides in heart muscle is recently under investigation and considered to be unfavorable process [33,34,35], however, which might be modified [36,37]. In this respect and with respect to findings in other tissues, the favorable effect of estradiol in this case is questionable and it seems that estradiol could cause also unfavorable shifts of fat content between different body compartments [37], including myocardial tissue. Therefore, the evaluation also of other organs and tissues in our study was important. Interesting finding in this respect was the changes of adiposity index (Table 1) as the marker for central fat; it was not affected by ovariectomy but decreased when ovariectomy was followed by estradiol treatment. Therefore, it followed reverse pattern as observed in the content of triglycerides in myocardial tissue (Figure 3); IL-6 could also modify this process as discussed later. In addition, also changes in the liver and renal cortex reflected significant increase in the content of triglycerides after ovariectomy reversed when ovariectomy was followed by estradiol substitution (Table 1). Notably, these rather robust changes in the mass and content of organs and tissues were not accompanied by changes in circulating FFA, and cholesterol, non-fasting glycemia, circulating insulin and glucagon. The explanation could be from previous experimental study in Wistar rats [38] indicating that organ changes after ovariectomy could precede changes in circulating factors, in this case not only changes of circulating lipids but also of parameters associated with glucose homeostasis. Regarding changes in circulating serum lipids, triglycerides decreased after ovariectomy and increased when estradiol was substituted after ovariectomy. It means that changes of serum triglycerides were in opposite direction compared to changes of the content of triglycerides in the liver and in the renal cortex (Table 1 and Table 2). Therefore, changing levels of estradiol were associated with redistribution between serum triglycerides and triglycerides in these organs. In addition, concentration of HDL cholesterol increased significantly after ovariectomy and increased even further if ovariectomy was followed by estradiol substitution. It should be noted, that neither in human plasma serum/plasma concentration of HDL cholesterol exactly reflect structure or mechanisms of the real metabolic activity of HDL particles [39]. In addition, in rodents HDL cholesterol could play different roles than in humans [40]. Therefore, changes of HDL cholesterol concentrations after ovariectomy and ovariectomy followed by estradiol treatment definitely could not be definitely defined as favorable, i.e., concentration of HDL cholesterol does not reflect only positive effects. Notably, HDL cholesterol could interact also with non-cardiovascular structures [41] and also changes of HDL cholesterol concentration caused by ovariectomy and/or estradiol treatment are rather complex [42].

The last interesting finding was the change of supposedly pro-inflammatory factor, circulating IL-6 (Table 2). These changes were of similar pattern as changes in aortic vascular strain, myocardial, liver and kidney mass and of reverse pattern than was observed in adiposity index. It means IL-6 significantly decreased after ovariectomy and increased to concentration significantly higher than in the sham group if ovariectomy was followed by estradiol substitution. Such change of IL-6 seems to be counterintuitive, but IL-6 in the early stages of vascular impairment could play more ambiguous and even protective role on the vessel wall [43]. From previous findings, the role of the IL-6 is really ambiguous and could exert different effects on cardiovascular system especially in the terrain of fluctuating sex hormones [44,45]. Additionally, it cannot be excluded that IL-6 could be involved also in the shift of lipids between various organs and tissues including myocardial muscle [46,47].

Limitation of our study is the absence of blood pressure values at the time of ultrasound measurements of aortic strain; blood pressure at the time of measurements is very tightly associated with this parameter. Nevertheless, invasive methods, the most reliable methods for blood pressure measurements in experimental models to date, can adversely affect obtained results; changes in aortic strain were logical and offered biological plausibility. In addition, in our study we were focused not only on the arterial properties but also on cardiac and non-cardiac changes. Another limitation could be the focus on the abdominal aorta only and not on the other vessels, experimental studies in this area are focused mainly on the model of abdominal aneurysm [48]. The evaluation of the whole vasculature could be now possible with ultrasound studies. 

Another potential limitation to be discussed is the dosage of estradiol and its relatively high concentration after substitution. In our study, the concentration of estradiol after supplementation was much higher than in the sham group (315 ± 57 pg/mL vs. 35 pg/mL). In this respect, the dosage of estradiol was based on the information from previous studies and was similar as in other published experiments. Dosage of estradiol 3 μg/day administered through pump device, produced concentration of estradiol after 6 weeks in serum 242 ± 89 pg/mL [49]. Polito et al. [50] administered doses 0.003 and 0.03 mg of estradiol/kg per day in their experimental study without determined consequent estradiol levels in circulation. In general, in available literature, in rat females is the concentration of estradiol dependent on estrus and fluctuates in the range of 145–2100 pg/mL. In some papers it could fluctuate in the range of 20–60 pg/mL. From vascular point of view, higher doses of estradiol were studied intensively in the cerebral circulation and, interestingly, proved to be protective irrespectively of ovariectomy [51]. However, the effects of estradiol could be also dependent on the route of administration [52].

The strength of our study is specific design of repeated ultrasound measurements of vascular changes in the same animal before and after intervention focused on functional arterial properties and, more importantly complex assessment not only of cardiovascular but also extracardiac parameters with possibility to study potential interactions between cardiac, vascular and other organs and tissues after ovariectomy and estradiol substitution in the terrain of insulin resistance. Another advantage of this study is the focus on reciprocal interactions between multiple organs after mimicking menopause and hormonal substitutional therapy; as to our best knowledge, such approach was very rarely applied.

The main message from this study is that not all cardiovascular changes after estradiol substitution after ovariectomy need to be favorable especially at the background of already present metabolic disorders including insulin resistance accompanied by the inflammatory status. These findings highlight need to assess changes after menopause and hormonal substitution therapy in a complex manner including also assessment of the role of extravascular and extracardiac structures and the complex status of organism at the time of rapid hormonal changes induced by menopause but also by sex hormone substitution. In another words, effects of estradiol substitution after menopause really need not to be favorable in all aspects and in all individuals.

## 4. Materials and Methods

### 4.1. Animals and Diet

Design/flowchart of the study is presented in Figure 4. HHTg female rats were included in the study as optimal model for prediabetes and insulin resistance [13]. This strain is characterized by the presence of genetically determined hypertriglyceridemia, insulin resistance of peripheral tissues and hepatic steatosis but with the absence of obesity and fasting hyperglycemia. All animals used in the present study were bred at the animal house of the Center of Experimental Medicine, Institute for Clinical and Experimental Medicine (IKEM, Prague, Czech Republic). All of the experiments were performed in agreement with the Animal Protection Law of the Czech Republic (311/1997), which is in compliance with European Community Council recommendations (86/609/ECC) for the use of laboratory animals, and were approved by the Ethics Committee of the Institute for Clinical and Experimental Medicine, Project No. 6/2020. Rats were kept at a temperature of 22 °C and humidity-controlled conditions under a 12/12 h light/dark cycle with free access to a standard chow diet (Altromin, Maintenance diet for rats and mice, Lage, Germany) and drinking water. At the beginning of the study female rats were randomly divided into three experimental groups (*n* = 6), with measurements taken for body weight, serum glucose and triglycerides. At 8 weeks of age, were anesthetized with ketamine (70 mg/kg) and xylazine (10 mg/kg) administered intraperitoneally and then bilaterally ovariectomized using a midline incision (W-OVX). Sham-operated animals underwent the entire surgery, except for the removal of ovaries. Animals were saturated with oxygen throughout the procedure followed by subcutaneous analgesia (meloxicam 1 mg/kg). The health status of animals was monitored post-surgery. Two weeks after ovariectomy, 17-β estradiol subcutaneous therapy, in a dose 12.5 μg of 17-β estradiol (Sigma-Aldrich, St. Louis, MO, USA)/ kg body weight per day, was started in HHTg group supplemented with estradiol after ovariectomy. Food intake was measured weekly over a 4-month period to ensure the likely development of metabolic disorders associated with postmenopausal metabolic syndrome, as reported in previous studies [53]. At the end of the experiment, rats were sacrificed by decapitation after light anaesthetization (zoletil 5 mg/kg b.wt.) in a postprandial state. Aliquots of serum and tissue samples were collected and stored at −80 °C for further analysis.

### 4.2. Ultrasound Studies

Animal experiments were performed under a protocol approved by the Committee of the Institute. Rats were examined by a high-resolution US imaging system (Vevo 2100, FUJIFILM VisualSonics Inc., Toronto, Canada). The rats were anesthetized with isoflurane using an induction chamber connected with a scavenger canister. After induction, each animal was placed on a temperature-controlled board, and the four limbs coated with conductive paste and taped on the ECG electrodes. During the examination, the animals were maintained under gaseous anesthesia by a nose cone (1.5–2% isoflurane in 0.4–0.8 L/min in pure oxygen) and heart rate, respiration frequency and electrocardiogram were monitored. The abdomen was shaved and coated with acoustic coupling gel. M-mode images of abdominal aorta directly above and under the origin of renal arteries were obtained with Vevo 2100 high-resolution in vivo microimaging system using MS 250S transducer (20 MHz) (FUJIFILM VisualSonics Inc., Toronto, Canada) held in position by a hand and/or mechanical arm was used for the image acquisitions. B-mode images were obtained placing the US probe above the abdominal aorta to obtain cross-sectional images with the region of interest located in the focal zone of the transducer. Screening of aorta including aortic dilation was assessed by ultrasound 5 days prior surgery and 14 weeks after surgery (i.e., after 12 weeks of hormone replacement therapy) as a surrogate marker of aortic elasticity. Examiners and readers were blinded to the status of animals. Three consecutive cycles (maximal and minimal aortic diameter) were measured in three separate recordings for both, abdominal aorta above and under the origin of renal arteries. Aortic dilation was calculated as a difference between the averages of maximal and minimal aortic diameters gated by ECG in systole and diastole. Data were evaluated in a blinded fashion. Strain in abdominal aorta was calculated as previously described in carotid arteries [54,55,56]. In short, the strain was expressed as percent change in the arterial diameter: strain = (SD − DD)/DD, where SD was the systolic and DD the diastolic aortic diameter. The main principle of the measurement of abdominal dilation used for aortic strain calculation is shown in Figure 5.

### 4.3. Analytical Methods and Biochemical Analyses

Serum levels of triglycerides (Kit number: TG250, BLT00059), glucose (Kit number: GOD 1500, 132410), total and HDL cholesterol (Kit number: CHOL250, BLT00036 and HDLC4, 07528566), ALT (Kit number: 94973UN18), AST (Kit number: 73201UN19), and FFA (Kit number: half micro test, 11383175001) were measured using commercially available kits (Erba Lachema, Brno, Czech Republic, and Roche Diagnostics, Mannheim, Germany). Serum insulin (Kit number: 10-1250-01), glucagon (Kit number: YK90), IL-6 (Kit number: MBS 701221), AMH (Kit number: MBS 264077) and NOS (Kit number: MBS 261741) concentrations were determined using the rat ELISA kit (Mercodia AB, Uppsala, Sweden; Yanaihara Institute Inc., Fujinomiya-shi, Japan, MyBioSource, San Diego, CA, USA). Serum 17β-estradiol (Kit number: DLS 4800) and 17β-hydroxyprogesterone (Kit number: IM 1452) were analyzed using rat RIA kits (Immunotech, Prague, Czech Republic). To determine triglyceride and cholesterol content in tissues, samples were extracted using a chloroform/methanol mixture. The resulting pellet was dissolved in isopropyl alcohol, with triglyceride content determined by enzymatic assay (Erba-Lachema, Brno, Czech Republic). Perimetrial fat pads (Figure 6) were removed and weighed to assess adiposity index according to already established methodology [57] and the adiposity was determined by the adiposity index (the sum of the weight of perimetrial inguinal and perimetrial white adipose tissue divided by body weight).

### 4.4. Gene Expression

Total RNA were isolated from abdominal aorta and myocardial tissue using RNA Blue (Top-Bio, Vestec, Czech Republic). Reverse transcription and quantitative real-time PCR analysis was performed using the TaqMan RNA-to-C_T_ 1-Step Kit and TaqMan Gene Expression Assay (Applied Biosystems, Waltham, MA, USA). Relative expressions of *Nos3* (Assay ID: Rn02132634_s1) and *Cx37* (Assay ID: Rn00572193s1) were determined after normalization against *Hprt* gene as an internal reference and calculated using 2-ΔΔCt method, with results run in triplicate. 

### 4.5. Statistical Analyses

Data are given as the mean ± SD. One-way ANOVA was applied to analyze the parameters describing the differences between variables before and after ovariectomy and before and after estradiol treatment and Fisher LSD post-hoc test was used. When comparing difference in suprarenal aortic strain of the group after ovariectomy and estradiol substitution with the sham group, unpaired Student’s t-test was used. Statistical significance was defined as *p* < 0.05. Statistical analysis was performed using BMDP Statistical Software.

## 5. Conclusions

Ovariectomy in HHTg rats caused robust changes of the properties of abdominal aorta and also of the mass and the structure of the heart. Most vascular and cardiac changes after ovariectomy were reversed by estradiol treatment and some parameters reached values even different from sham group. In addition, some parameters were changed by estradiol supplementation irrespectively of ovariectomy. Moreover, vascular and cardiac changes were accompanied by similar or reverse changes in other organs and tissues including central adiposity, liver and kidney. In summary, these findings indicate different effects of ovariectomy and estradiol on different cardiovascular but also extravascular and extracardiac systems and structures and should be taken into account and studied in parallel. 

## Figures and Tables

**Figure 1 ijms-23-02825-f001:**
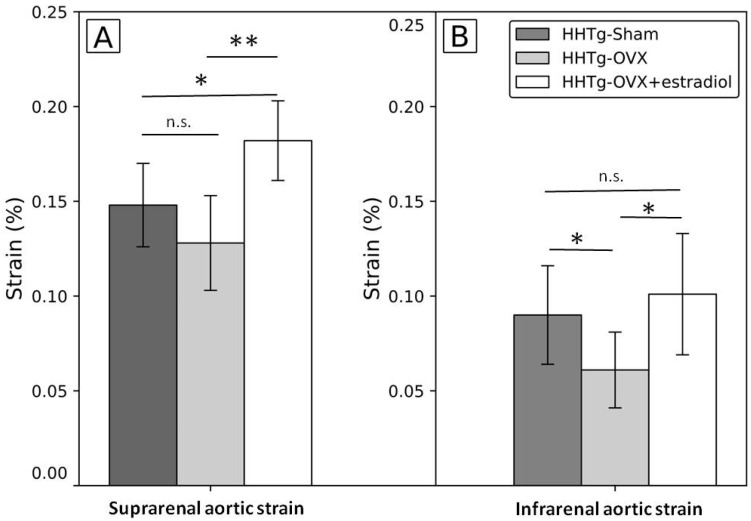
Differences in the strain of the abdominal aorta after ovariectomy and estradiol substitution in HHTg rats in suprarenal (**A**) and infrarenal (**B**) segments. Legend: Strain: (systolic-diastolic aortic diameter)/diastolic aortic diameter; HHTg: hereditary hypetriglyceridemic; HHTg-Sham: sham group, *n* = 5; HHTg-OVX: Ovx: ovariectomy (*n* = 6); HHTg-OVX + estradiol: ovariectomy followed by estradiol substitution (*n* = 6). Data are expressed as mean ± SD and evaluated by one-way ANOVA and Fisher LSD post-hoc test; n.s.: nonsignificant, * *p* < 0.05, ** *p* < 0.01.

**Figure 2 ijms-23-02825-f002:**
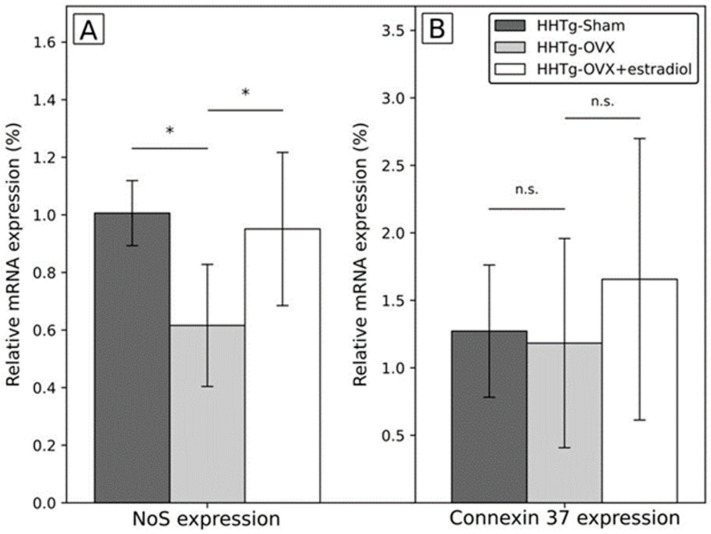
Differences in the expression of genes in abdominal aorta after ovariectomy and estradiol substitution in HHTg rats. Legend: HHTg: hereditary hypetriglyceridemic; HHTg-Sham: sham group, *n* = 5; HHTg-OVX: Ovx: ovariectomy (*n* = 6); HHTg-OVX + estradiol: ovariectomy followed by estradiol substitution (*n* = 6). (**A**) NoS expression: expression of the gene for *nitric oxid synthase*, (**B**) Connexin 37 expression: expression of the gene for *connexin 37*. Data are expressed as mean ± SD and evaluated by one-way ANOVA and Fisher LSD post-hoc test; n.s.: nonsignificant, * *p* < 0.05.

**Figure 3 ijms-23-02825-f003:**
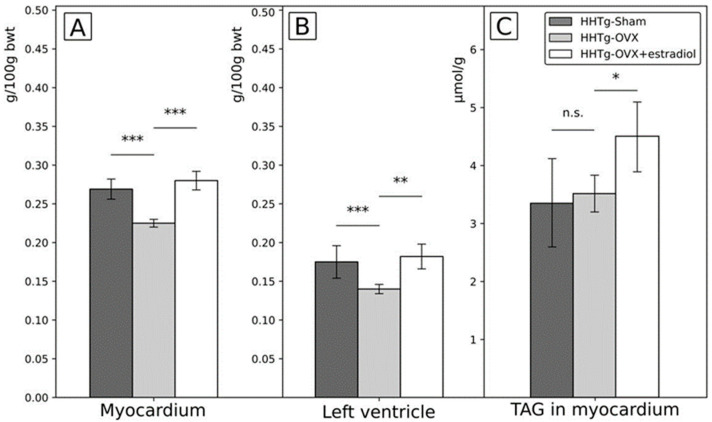
Differences in the relative weight of myocardium (**A**) and the left ventricle (**B)** and triglycerides content of the myocardium (**C**) after ovariectomy and estradiol substitution in HHTg rats. Legend: HHTg: hereditary hypetriglyceridemic; HHTg-Sham: sham group, *n* = 5; HHTg-OVX: Ovx: ovariectomy (*n* = 6); HHTg-OVX + estradiol: ovariectomy followed by estradiol substitution (*n* = 6). Data are expressed as mean ± SD and evaluated by one-way ANOVA and Fisher LSD post-hoc test; n.s.: nonsignificant, * *p* < 0.05, ** *p* < 0.01, *** *p* < 0.001.

**Figure 4 ijms-23-02825-f004:**
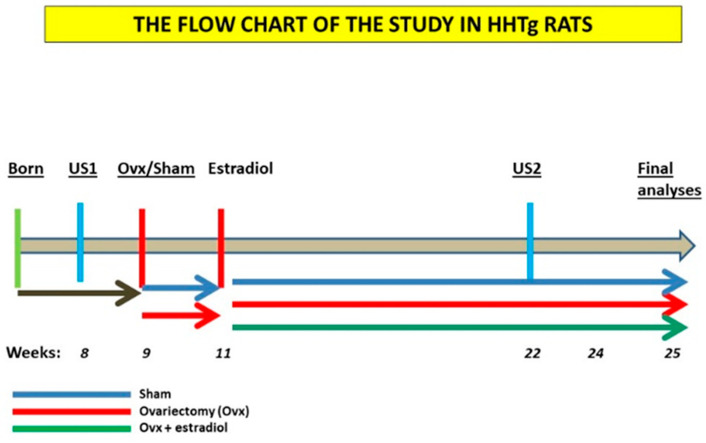
Flow chart of the experiment. HHTg: hereditary hypetriglycerdiemic, US1—first ultrasound examination of abdominal aorta, US2—second ultrasound examination of abdominal aorta, Ovx: ovariectomy.

**Figure 5 ijms-23-02825-f005:**
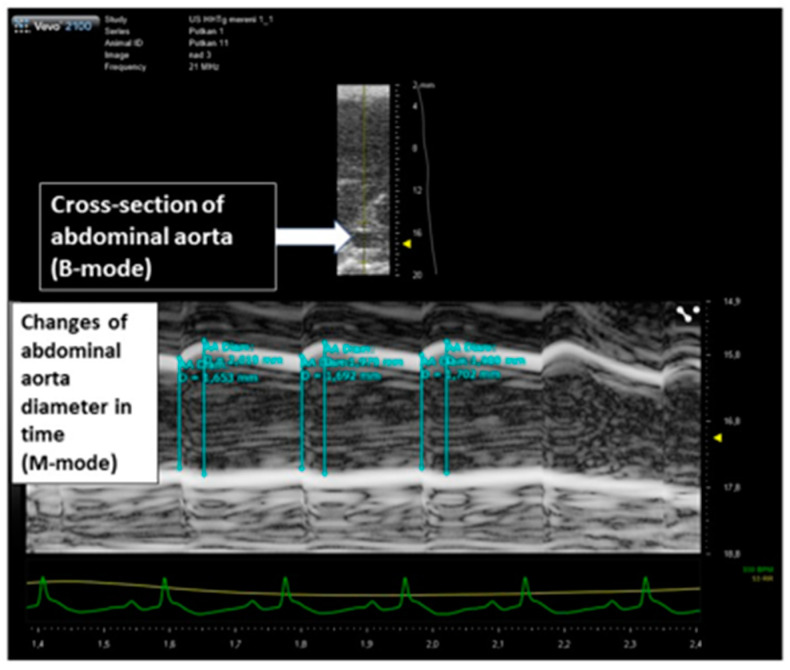
The principle of the assessment of abdominal aortic dilation in HHtg rats (ultrasound/M-mode measurements) for aortic strain calculation.

**Figure 6 ijms-23-02825-f006:**
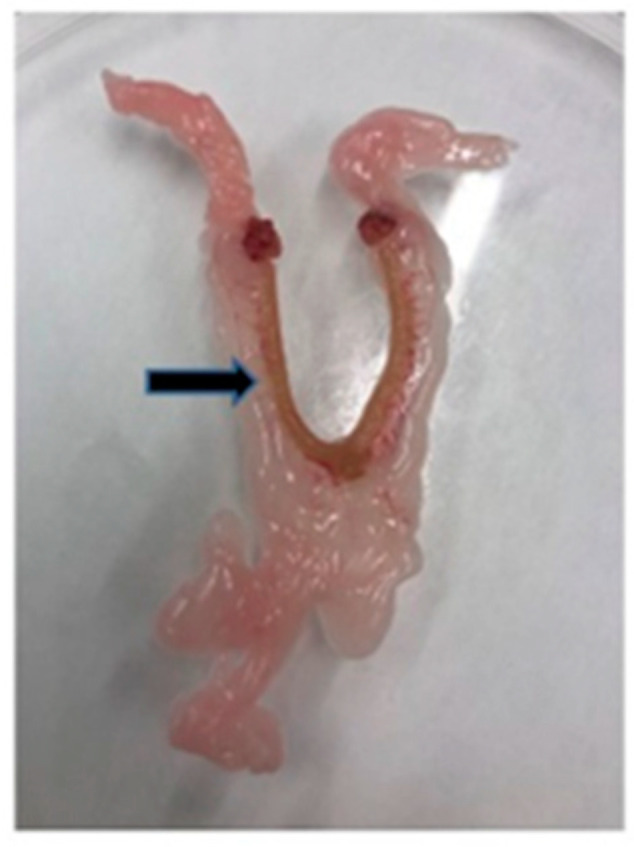
Perimetrial fat (black arrow) obtained for evaluation of the adiposity index (rat uterus).

**Table 1 ijms-23-02825-t001:** Differences after Ovariectomy and Estradiol Substitution in Hereditary Hypertriglyceridemic Rats in the Relative Weight and Lipid Content of Extracardiac Tissues and Organs.

	Sham *n* = 5	Ovx *n* = 6	Ovx + Estradiol *n* = 6	One-Way ANOVA	*p* Ovx vs. Sham	*p* Ovx + E vs. Ovx
Body weight (g)	242.4 ± 8.2	272.2 ± 12.9	238.0 ± 12.9	0.001	0.001	0.001
Uterus (g/100 g bwt)	0.226 ± 0.029	0.098 ± 0.034	0.290 ± 0.069	0.001	0.001	0.001
Adiposity index (g/100 g bwt)	2.051 ± 0.453	2.253 ± 0.397	1.620 ± 0.336	0.05	n.s.	0.05
Liver (g/100 g bwt)	3.283 ± 0.095	2.556 ± 0.042	3.769 ± 0.094	0.001	0.001	0.001
Hepatic content of triglycerides (μmol/g)	7.885 ± 1.853	12.970 ± 1.533	10.215 ± 0.851	0.001	0.001	0.01
Hepatic content of cholesterol (μmol/g)	6.581 ± 0.826	9.950 ± 0.593	6.763 ± 1.029	0.001	0.001	0.001
Kidneys (g/100 g bwt)	0.570 ± 0.015	0.438 ± 0.009	0.620 ± 0.027	0.001	0.001	0.001
Content of triglycerides in the renal cortex (μmol/g)	0.771 ± 0.067	1.075 ± 0.112	0.824 ± 0.049	0.001	0.001	0.001
Content of triglycerides in the skeletal muscles (μmol/g)	1.223 ± 0.242	1.467 ± 0.205	1.491 ± 0.621	n.s.	n.s.	n.s.

Legend: Data are given as the mean ± SD. One-way ANOVA and Fisher LSD post-hoc test were used. Ovx: ovariectomy, Ovx + E: Ovariectomy and estradiol substitution. Data are expressed as mean ± SD and evaluated by one-way ANOVA and Fisher LSD post-hoc test; n.s.: non.significant.

**Table 2 ijms-23-02825-t002:** Differences after Ovariectomy and Estradiol Substitution in Circulating Metabolic and Inflammatory Parameters in Serum in Hereditary Hypertriglyceridemic Rats.

	Sham *n* = 5	Ovx *n* = 6	Ovx + Estradiol *n* = 6	One-Way ANOVA	*p* Ovx vs. Sham	*p* Ovx + E vs. Ovx
17β-estradiol (pg/mL)	35.16 ± 4.40	23.37 ± 3.58	314.95 ± 104.33	0.001	n.s.	0.001
Progesterone (ng/mL)	1.542 ± 0.366	0.417 ± 0.083	0.563 ± 0.061	0.001	0.001	n.s.
Anti-Mullerian hormone (ng/mL)	6.725 ± 1.645	6.654 ± 2.201	5.884 ± 0.636	n.s.	n.s.	n.s.
Cholesterol (mmol/L)	1.564 ± 0.265	1.795 ± 0.181	2.027 ± 0.244	0.01	n.s.	n.s.
Triglycerides (mmol/L)	4.826 ± 1.073	2.220 ± 0.706	4.187 ± 0.698	0.001	0.001	0.01
HDL-cholesterol (mmol/L)	0.808 ± 0.085	1.025 ± 0.138	1.230 ± 0.078	0.001	0.01	0.01
Free fatty acids (mmol/L)	0.520 ± 0.132	0.625 ± 0.109	0.528 ± 0.091	n.s.	n.s.	n.s.
Non-fasting glucose (mmol/L)	8.260 ± 0.666	8.350 ± 0.709	8.317 ± 0.741	n.s.	n.s.	n.s.
Insulin (nmol/L)	0.191 ± 0.059	0.160 ± 0.034	0.151 ± 0.030	n.s.	n.s.	n.s.
Glucagon (pg/mL)	201.8 ± 25.5	214.1 ± 37.1	219.2 ± 18.6	n.s.	n.s.	n.s.
Alanine aminotransferase (μkat/L)	0.960 ± 0.082	1.250 ± 0.096	1.222 ± 0.118	0.001	0.001	n.s.
Aspartate aminotransferase, (μkat/L)	2.540 ± 0.179	3.202 ± 0.309	2.977 ± 0.226	0.01	0.001	n.s.
Circulating Interleukin 6 (pg/mL)	106.65 ± 7.47	72.73 ± 14.98	173.31 ± 22.06	0.001	0.01	0.001

Legend: Data are given as the mean ± SD, One-way ANOVA and Fisher LSD post-hoc test were used. Ovx: ovariectomy, Ovx + E: Ovariectomy and estradiol substitution. Data are expressed as mean ± SD and evaluated by one-way ANOVA and Fisher LSD post-hoc test; n.s.: nonsignificant.

## Data Availability

All datasets generated for this study are included in the article.

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
