# Peer review of "Cardiovascular, Metabolic and Inflammatory Changes after Ovariectomy and Estradiol Substitution in Hereditary Hypertriglyceridemic Rats"

_ijms, 2022, doi:10.3390/ijms23052825_

Round 1

Reviewer 1 Report

The authors have mostly addressed my previous comments. One typo to be corrected: in Table 1 & 2, Ovx+E group numbers are "-6".

Author Response

The authors have mostly addressed my previous comments. One typo to be corrected: in Table 1 & 2, Ovx+E group numbers are "-6

Thank you - corrected.

Reviewer 2 Report

The authors examined the cardiovascular, metabolic and inflammatory changes after ovariectomy and estradiol replacement in hereditary. They found that some of the changes after ovariectomy were reversed by estradiol replacement and some are not. The study provided some insights into hormone replacement for cardiovascular and metabolic disorders in postmenopausal women.

Here are some concerns about this paper:

Page 3, line 12

"the strain after estradiol substitution was higher not only than in ovariectomized rats (p=0.002) but also than in the sham group (p=0.02) (Figure 1, Table A1) "

Significance should be shown in Fig. 1.

Page 8, line 22

"Nevertheless, it should be noted, that changes after ovariectomy/estradiol substitution both in cx37 gene expression, but also in circulating NOS were very similar to changes of aortic strain and of Nos3 but with higher interindividual variability leading to the absence of statistical significance (Figure 2, Table A1). "

This discussion is nonsense; it is not based on the data and should be removed, or should show data appropriately.

Page 8, line 14 from the bottom

“Changes levels of estradiol could be also the cause of the significant changes of the heart rate, this could be caused also by changes in estradiol receptors and their impact on sensitivity of the heart to stress.”

Were ERs in the myocardium downregulated after ovariecotmy? Need reference citation.

Page 9, line 26

“These changes closely and inversely mirrored the changes of the content of triglycerides in the liver and in the renal cortex; therefore, redistribution between plasma triglycerides and triglycerides in these organs was clearly associated with changing levels of estradiol.”

Why do the authors consider “closely and inversely” mirrored? Is there a statistical basis?

Page 11, 4.2. Ultrasound studies.

The strain of the vessels is very small (<0.25%), and therefore, very fine measurement seems to be required. Please show the resolution of the measurement, and if possible the typical images.

Author Response

The authors examined the cardiovascular, metabolic and inflammatory changes after ovariectomy and estradiol replacement in hereditary. They found that some of the changes after ovariectomy were reversed by estradiol replacement and some are not. The study provided some insights into hormone replacement for cardiovascular and metabolic disorders in postmenopausal women.

Here are some concerns about this paper:

Page 3, line 12

"the strain after estradiol substitution was higher not only than in ovariectomized rats (p=0.002) but also than in the sham group (p=0.02) (Figure 1, Table A1) " Significance should be shown in Fig. 1.

Thank you, we added statistical significances to be complete in Fig 1.

 Page 8, line 22

"Nevertheless, it should be noted, that changes after ovariectomy/estradiol substitution both in cx37 gene expression, but also in circulating NOS were very similar to changes of aortic strain and of Nos3 but with higher interindividual variability leading to the absence of statistical significance (Figure 2, Table A1). " This discussion is nonsense; it is not based on the data and should be removed, or should show data appropriately.

Thank you for this comment. We did our best to address this comment - the point was, that really something could be happening there. To avoid confusion we changed/soften this statement and excluded ref. No 27.

Page 8, line 14 from the bottom

“Changes levels of estradiol could be also the cause of the significant changes of the heart rate, this could be caused also by changes in estradiol receptors and their impact on sensitivity of the heart to stress.” Were ERs in the myocardium downregulated after ovariecotmy? Need reference citation.

Thank you for this comment. In previous discussion/statement we were already cautious about ER/estradiol mechanisms and changes in myocardium. We further modified these statements and provided other two citations - some data were already provided in ref. (now) 28 plus we added references 29-30. In summary, we are not able to prove changes of number estradiol receptors - definitely not from this study. From available data rather the function of ER could be changed. But, based on ongoing discussion in this field, we added one more sentence: “ …it should be noted, that estradiol can have discordant cardiac effects based on the experimental models used.“  Pls. see our revisions.

Page 9, line 26

“These changes closely and inversely mirrored the changes of the content of triglycerides in the liver and in the renal cortex; therefore, redistribution between plasma triglycerides and triglycerides in these organs was clearly associated with changing levels of estradiol.”Why do the authors consider “closely and inversely” mirrored? Is there a statistical basis?

Thank you for this comment. We tried to improve clarity of this statement.

 Page 11, 4.2. Ultrasound studies.

The strain of the vessels is very small (<0.25%), and therefore, very fine measurement seems to be required. Please show the resolution of the measurement, and if possible the typical images.

We definitely agree that especially in ultrasound studies (event with axial resolution less than 0,1 mm) focused on subtle changes the variability could be very important factor. Unfortunately because of design of the study we were not able to provide repeated studies in vivo regarding variability. Nevertheless, we believe that using mean of 3 subsequent measurements could substantially diminish this variability and blinding of examiners/readers guaranteed reliability. We also added typical image - Fig. 5 (it was included in on of previous versions, but removed based on one of previous recommendation from reviewers).

This manuscript is a resubmission of an earlier submission. The following is a list of the peer review reports and author responses from that submission.

Round 1

Reviewer 1 Report

This manuscript by Pitha et al. investigated if ovariectomy in female rats contribute to increased cardiovascular risks and if estradiol substitution can correct these changes. This study addressed an important scientific question that remained to be answered. Also, the merits of this study include the innovative approach using ultrasound and the comprehensive design using a rat model with metabolic disorder background. The findings supported the hypothesis that menopause can increase risks for cardiovascular diseases in metabolically disturbed populations and hormone substitution therapy could potentially reverse some of these problems. Overall, it is a research with high value, although some language and graphic presentation need to be corrected.

  1. The authors tend to use ambiguous descriptions throughout the text: for example, “negative vascular changes”, “favorable vascular effects”, “cardiovascular changes”, “atherosclerotic changes” were used in the first paragraph in introduction, without any clarification. The authors need to state clearly what are the key changes of increased atherosclerotic risks in arteries, and what are considered improvement or protection against these risks.
  2. The figures require major improvement. These is no y-axis label. It is unclear what the error bars stand for (e.g. mean+/- stdev or sem). The asterisk marks are out of place and not defined in the figure legend. Number per group needs to be added to legend. Two panels in figure 3 should be the same size.
  3. Similarly it is unclear what were compared for the P values in tables. Clearly state whether Ovx+E was compared to control or Ovx.

Author Response

  1. The authors tend to use ambiguous descriptions throughout the text: for example, “negative vascular changes”, “favorable vascular effects”, “cardiovascular changes”, “atherosclerotic changes” were used in the first paragraph in introduction, without any clarification. The authors need to state clearly what are the key changes of increased atherosclerotic risks in arteries, and what are considered improvement or protection against these risks.

Thank you for the comment and proposal. We did our best to standardize and unify terminology and make the first part of the Introduction more clear.  

  1. The figures require major improvement. These is no y-axis label. It is unclear what the error bars stand for (e.g. mean+/- stdev or sem). The asterisk marks are out of place and not defined in the figure legend. Number per group needs to be added to legend. Two panels in figure 3 should be the same size.

We apologize for this technical mistake and we have redone the Figures to be more clear and informative, in addition, the Figure(s) 3 a/b were merged in one.

  1. Similarly it is unclear what were compared for the P values in tables. Clearly state whether Ovx+E was compared to control or Ovx.

We expanded the Legend of all Tables and made Headings clearer with regard to statistical evaluation.

Reviewer 2 Report

The manuscript "Cardiovascular, Metabolic and Inflammatory Changes after Ovariectomy and Estradiol Substitution in Hereditary Hypertriglyceridemic Rats" is interesting. Following modifications are required to improve the manuscript quality.

Comments

In this study, did you noticed any correlation between Nos3 and cx37?

Include heart rate and electrocardiogram data.

In all graphs, the Y-axis title is missing.

In figure legends, explain the p-value, value of "n", and explain how the values are expressed (mean+/-SD).

In methodology, provide details of kits (catalog number).

Author Response

Comments and Suggestions for Authors

The manuscript "Cardiovascular, Metabolic and Inflammatory Changes after Ovariectomy and Estradiol Substitution in Hereditary Hypertriglyceridemic Rats" is interesting. Following modifications are required to improve the manuscript quality.

Thank you for your comments which we consider as relevant, please, see our answers.

Comments

In this study, did you noticed any correlation between Nos3 and cx37?

We evaluated this and add into the text (). However, as also mentioned in Discussion, because of low number of animals in each group we are cautious to make clear conclusion based on these data. .

Include heart rate and electrocardiogram data.

We included these data and analyzed them - please see the text, new Figure 5 and Figure A1.. Surprisingly there were found some interesting differences, but again because of rapid changes in the heart rate during ultrasound procedures and relatively low number of animals, we made rather cautious conclusions. .

In all graphs, the Y-axis title is missing. We apologize for this technical mistake which we corrected. We have redone the Figures to be more clear and informative, in addition, the Figure 3 was merged in one.

In figure legends, explain the p-value, value of "n", and explain how the values are expressed (mean+/-SD). It was corrected.

In methodology, provide details of kits (catalog number).

The kit numbers were added.

Round 2

Reviewer 2 Report

The authors revised the manuscript "Cardiovascular, Metabolic and Inflammatory Changes after Ovariectomy and Estradiol Substitution in Hereditary Hypertriglyceridemic Rats," but further modification is required to improve the manuscript quality.

Figure 5 does not look self-explanatory, and I think it would be better to remove figure 5.

Double-check and include the Y-axis title in all graphs and figure numbers like Figure 1A, 1B, etc.

The animal number N=6 is not relatively low. Explain the cause of variability in heart rate in the discussion part. Include the value of N in all figure legends.

Additionally, change the figure number for Figure A1(Figure 5) and correct the figure legends.

Author Response

The authors revised the manuscript "Cardiovascular, Metabolic and Inflammatory Changes after Ovariectomy and Estradiol Substitution in Hereditary Hypertriglyceridemic Rats," but further modification is required to improve the manuscript quality.

Thank you for your additional comments/proposals, please, see below and changes in the text/figures.

Figure 5 does not look self-explanatory, and I think it would be better to remove figure 5.

We removed it.

Double-check and include the Y-axis title in all graphs and figure numbers like Figure 1A, 1B, etc.

We did improve the graphs and added y axis, renumbered Figures.

The animal number N=6 is not relatively low. Explain the cause of variability in heart rate in the discussion part. Include the value of N in all figure legends.

We did our best to discuss these results, pls., see changes, because of different values at the beginning of the study were are still cautious to stress this finding as of critical importance.

Additionally, change the figure number for Figure A1(Figure 5) and correct the figure legends.

We did.

Round 3

Reviewer 2 Report

The manuscript "Cardiovascular, Metabolic and Inflammatory Changes after Ovariectomy and Estradiol Substitution in Hereditary Hypertriglyceridemic Rats" is acceptable for publication.